# Allelic decomposition and exact genotyping of highly polymorphic and structurally variant genes

Ibrahim Numanagić [1,2,6], Salem Malikić[1], Michael Ford [1], Xiang Qin[3], Lorraine Toji[4], Milan Radovich[5], Todd C. Skaar[5], Victoria M. Pratt[5], Bonnie Berger [2,6], Steve Scherer[3] & S. Cenk Sahinalp[7]

High-throughput sequencing provides the means to determine the allelic decomposition for any gene of interest—the number of copies and the exact sequence content of each copy of a gene. Although many clinically and functionally important genes are highly polymorphic and have undergone structural alterations, no high-throughput sequencing data analysis tool has yet been designed to effectively solve the full allelic decomposition problem. Here we introduce a combinatorial optimization framework that successfully resolves this challenging problem, including for genes with structural alterations. We provide an associated computational tool Aldy that performs allelic decomposition of highly polymorphic, multi-copy genes through using whole or targeted genome sequencing data. For a large diverse sequencing data set, Aldy identifies multiple rare and novel alleles for several important pharmacogenes, significantly improving upon the accuracy and utility of current genotyping assays. As more data sets become available, we expect Aldy to become an essential component of genotyping toolkits.

[1] School of Computing Science, Simon Fraser University, Burnaby, BC V5A 1S6, Canada. [2] Computer Science and Artificial Intelligence Laboratory, Massachusetts Institute of Technology, Cambridge, MA 02139, USA. [3] Baylor College of Medicine Human Genome Sequencing Center, Houston, TX 77030, USA. [4] Coriell Institute for Medical Research, Camden, NJ 08103, USA. [5] Indiana University School of Medicine, Indianapolis, IN 46202, USA. [6] Department of Mathematics, Massachusetts Institute of Technology, Cambridge, MA 02139, USA. [7] Department of Computer Science, Indiana University, Bloomington, IN 47405, USA. Correspondence and requests for materials should be addressed to S.C.S. (email: cenksahi@indiana.edu)

Rapid development of high-throughput sequencing (HTS) technologies promises to resolve some of the most critical problems in genomics research and clinical genomic testing. In principle, HTS data provide the means to determine the exact sequence composition for each copy of any gene of interest. Unfortunately, from a computational point of view, this is a highly challenging task, since many functionally and clinically important genes are highly polymorphic and have multiple copies as well as sequencewise-similar pseudogenes, which they frequently hybridize/fuse with to produce novel alleles. In addition, some of these genes have been subject to structural alterations, making their allelic decomposition (i.e., determining the number of copies of a gene and the exact sequence content of each of its copies) computationally difficult.

Currently, no existing computational tool can utilize HTS data to perform allelic decomposition of genes that have been subject to structural alterations. Available structural variation detection tools aim to identify the type and locus of large "structure altering events" (e.g., large-scale deletions, novel sequence insertions, segmental duplications, and inversions), typically in uniquely mappable regions of the genome. In contrast, available copy number alteration detection/copy number phasing tools aim to identify the number of copies of a particular gene (in each chromosome[1]) under the implicit assumption that gene duplications or deletions always affect the entire (rather than a part of the) gene of interest, but do not reconstruct the exact sequence content of the gene (although the work of Sudmant et al.[1] does utilize small variants for copy number phasing, it is limited to detecting copy number changes only, as opposed to the work presented here, which can also determine the exact sequence content of each copy of a gene that has been subject to structural alterations). In fact, no existing tool aims to find out what happens when structural alterations affect genes with multiple copies or those with highly homologous pseudogenes. Such genes are algorithmically difficult to resolve since reads that originate from such genes have high mapping ambiguity.

In order to reconstruct the sequence content of a structurally altered gene, one needs to (i) find out how many copies of the gene there are and which read belongs to which copy (i.e., mapping ambiguity resolution), and (ii) implicitly or explicitly assemble each copy of the gene from the read set (this is inherently intermingled with mapping ambiguity resolution) and find out its origins (in the reference genome). This requires one to (a) identify all structural alteration breakpoints and carefully reconstruct the sequence content of each breakpoint region, while taking into account all micro-structural alterations, indels, and single nucleotide variants (SNVs) each copy of the gene has been subject to, and (b) identify fusions/hybridizations between the gene and its highly homologous pseudogenes. None of the computational tools available to date aim to address these issues.

Existing structural variation discovery tools are based on the following general strategies: detection of variants using discordantly mapping paired-end reads (e.g., VariationHunter[2,3] and HYDRA[4] which report only the rough loci of structural variants but not their sequence content); detection of variants using split-read mappings (e.g., Socrates[5]); and detection of variants by mapping de novo assembled contigs to a reference genome (e.g., Barnacle[6] and Dissect[7], which are RNA-Seq analysis tools that can also be used to analyze genomic data). There are also several tools that employ a combination of these (e.g., Pindel[8], Delly[9], novoBreak[10], and GASVPro[11]), which, unfortunately, only consider uniquely mapped reads and cannot identify alterations in repetitive DNA. In fact, no available tool, even those designed to identify gene fusions only (e.g., deFuse[12]), aims to reconstruct the sequence content of a fusion between a gene and a highly similar pseudogene. Additionally, no existing tool aims to infer variants from targeted capture sequencing data which are highly non-uniform in coverage (e.g., PGRNseq—see below for details). Even tools that aim to genotype a particular gene such as CYP2D6, namely Cypiripi[13] and Astrolabe (formerly Constellation)[14], respectively work only on uniform coverage sequencing data, or can determine the gene's sequence content only if it differs from the reference by SNVs but not structural variation.

In order to address the aforementioned computational challenges, we present the first framework to perform allelic decomposition of any gene of interest in HTS data. More specifically, our combinatorial framework can perform allelic decomposition of any gene that differs from the reference genome by (i) SNVs, (ii) short indels, (iii) full gene duplications or deletions (leading to copy number alteration), (iv) partial gene duplications or deletions, as well as (v) "balanced" fusions (i.e., those that preserve the structure of a gene) with highly homologous pseudogenes (the fusions can have one or more breakpoints). Our framework accomplishes this ambitious goal through a novel combinatorial optimization formulation which it uses to identify all possible combinations of genomic alterations and determine the sequence content of all copies of a gene in whole genome or targeted genome sequencing data. The practical implementation of this framework, which we named Aldy, has been used to genotype 10 of the most important pharmacogenes which regulate drug metabolism.

Note that much of clinical genotyping is still performed through targeted genotyping panels. These panels (e.g., Affymetrix DMET+ arrays and the Illumina ADME assays) are able to detect a common set of predefined variations and genotypes. However, rare or personal variants, while functionally significant, often cannot be captured by these panels. Unfortunately, rare variants of pharmacogenes (e.g., CYP2D6) can impact drug response[15]. As a result, new HTS-based targeted captures are rapidly being introduced to help identify novel variants in a cost-effective manner. A prime example is the PGRNseq capture panel[16], which targets 84 genes of pharmacogenomic interest (also known as ADME genes) that encode drug-metabolizing enzymes, drug transporters, and drug targets. For each of these genes, PGRNseq targets the exonic region and a few kilobases upstream and downstream of a gene's UTR region, covering more than 960 KB of the human genome through its first iteration (PGRNseq v.1). PGRNseq capture products are sequenced on the Illumina HiSeq platforms, providing low error rates while maintaining a very high depth of coverage (averaging 500× per chromosome) at a significantly lower cost than whole-genome sequencing (WGS). Even though PGRNseq (or WGS) data for some of the pharmacogenes are relatively straightforward to interpret, other, more difficult genes, such as CYP2D6, have proven difficult to analyze[16]. Such genes provide an excellent application area for Aldy.

On a large data set involving 96 cell lines sequenced via the (second iteration) PGRNseq v.2 protocol, comprised of 32 family trios, 137 cell lines sequenced with the PGRNseq v.1 protocol, and 25 WGS Illumina samples (from various sources), we show that Aldy is able to reconstruct the sequence content of each copy of some of the most challenging (structurally altered and polymorphic) genes in the human genome and identify many novel alleles, significantly improving the accuracy and utility of currently used genotyping assays. Moreover, Aldy is able to identify a large set of hybrid/fusion genes, composed of a coding gene and a highly similar pseudogene; such fusions are very difficult to detect with existing genotyping assays. Aldy has minimal impact on computational resources, and is capable of analyzing a high-coverage BAM file in less than a minute on a typical laptop computer.

## Results

**Overview of Aldy.** The primary goal of Aldy is to "reconstruct the structure and sequence content" of each copy of a particular gene present in the sample being analyzed. Following the well-established star-allele nomenclature in pharmacogenomics[17], we define a star-allele of a gene as a gene sequence which differs from the "wild type" (or canonical) gene sequence by a (non-empty) set of mutations. Thus, for our purposes, reconstructing the sequence content of a gene copy is identical to identification of the gene copy's star-allele, which could either be already known or possibly novel.

We distinguish two types of mutations and, as a consequence, star-alleles. We call any mutation that has an impact on the resulting protein product of the gene a gene-disrupting mutation (also known as functional mutations). These mutations include codon-changing single nucleotide polymorphisms (SNPs) and indels, as well as mutations outside the coding regions that affect the protein enzyme activity. Star-alleles which are defined solely by gene-disrupting mutations are called major star-alleles, and are assigned a unique number. For example, the canonical "wild type" star-allele is always assigned *1, while *2 describes a star-allele that harbors one or more gene-disrupting mutations compared to the *1. If a new major star-allele, that has not been reported in the literature, is discovered, it is a common practice to call it $*n\ell + 1$, where $\ell$ is the number of major star-alleles known up to that point[17]. Note that two major star-alleles can share a common mutation.

We call any mutation that does not impact the protein product a neutral mutation (also known as non-functional mutations). Any major star-allele can be extended with neutral mutations, and any such extension is called a minor star-allele. (If a copy of a gene includes only neutral mutations, then it is considered to be

an extension of the wild type star-allele.) In order to distinguish various minor star-alleles, a unique symbol (or, rarely, a pair of symbols) is attached to the major star-allele's number for each such extension. For example, minor star-allele *2A is formed by taking the set of gene-disrupting mutations for major *2 allele and extending it with some neutral mutations; *2B is formed in a similar manner, however the sets describing the neutral mutations of *2A and *2B are not identical (although the sets describing their gene-disrupting mutations are). If a new minor star-allele that is an extension to the star-allele *k is discovered, it is commonly called *kX where X is the lexicographically smallest letter which has not yet been used for minor alleles of *k[17].

Following the definitions above, we can rephrase Aldy's goal as follows: we aim to characterize the sequence composition of each copy of a gene present in the sample, which is by definition equivalent to inferring the major and minor star-allele label of such a gene copy. In case there is a need to define a new star-allele, we do so by minimizing the number of novel mutations and structural variations (the set of allowed structural variations are summarized below) that need to be added to or subtracted from a known star-allele to describe the new one. In order to achieve this goal, Aldy goes through the following steps for a given gene (see Fig. 1):

Read alignment and mutation detection where HTS reads are aligned to the reference genome and mutations present in target gene region are identified;

Copy number and structural variation estimation where the copy number of the gene is identified, and, if present, various structural variations are identified;

Major star-allele identification where the major star-allele of each gene copy is established; and

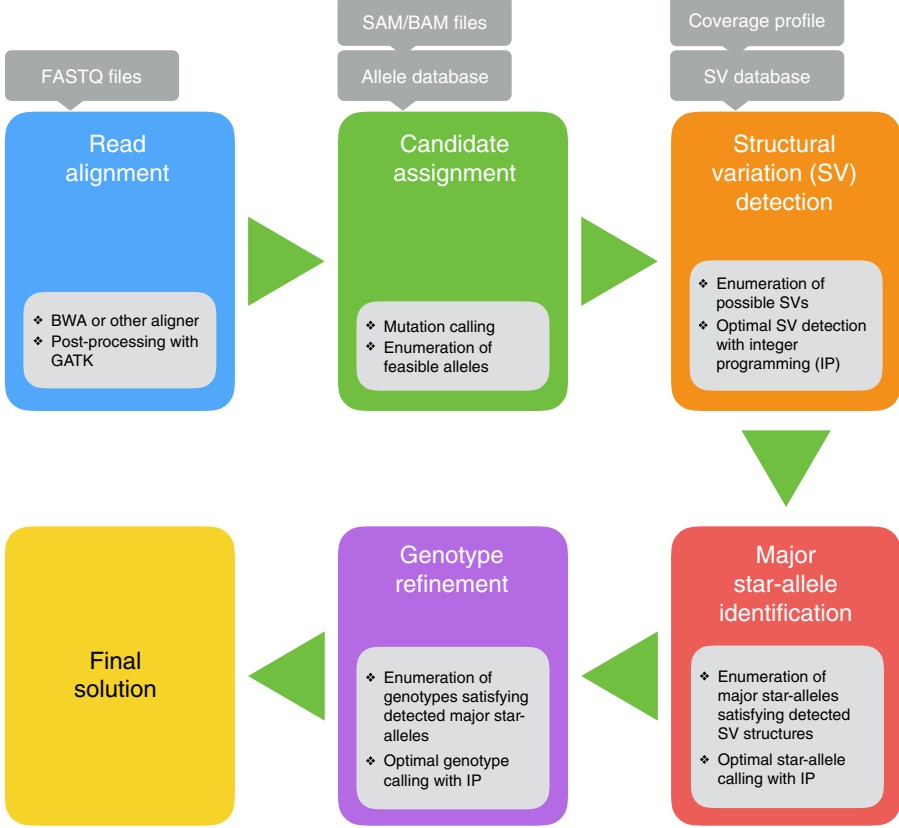

**Fig. 1** Graphical representation of steps performed by Aldy

Genotype refinement where the supporting set of neutral mutations is assigned to each major star-allele, and the "score" of such an assignment (see "Genotype refining" in Methods for details) is used to rank each allelic configuration found in the previous three steps.

The final genotype (i.e., minor star-allele) is obtained by choosing the set of allelic configurations with the best ranking score (as will be detailed in Methods). In the case of multiple configurations with the same score, all will be reported as equally likely genotypes.

The primary input to Aldy is HTS data in SAM/BAM file format, as well as a database comprised of information about the gene to be genotyped. Such a database will contain basic information about the gene (e.g., its location within a reference genome, locations of pseudogenes, intron/exon boundaries), as well as the list of all known major and minor star-alleles for that gene, each described as a unique set of gene-disrupting and neutral mutations. Furthermore, this database will also contain the list of all known structural variations involving the gene of interest; i.e., duplications and deletions, as well as all known hybridizations with its pseudogene, either in the form of fusions (when a prefix or suffix of the hybrid gene sequence is from the pseudogene) or gene conversions (when a segment other than a suffix/prefix of the hybrid gene is from the pseudogene).

Aldy uses the database to "guide" star-allele discovery. In other words, if possible, Aldy will aim to assign a known major and minor star-allele label for each copy of the gene. However, if no known star-allele description "matches" the input data, Aldy will infer previously unknown major or minor star-alleles, by minimizing the number of mutations or structural variations that need to be added to or subtracted from known star-allele descriptions. The mutations we consider here are SNVs and short indels; the structural variations we consider here are (partial) deletions or duplications of the gene, and hybridizations (a.k.a. fusions) with a specified pseudogene. In its current implementation, Aldy detects only hybrid gene structures that have been reported in the literature, or those that are manually entered in the database. For CYP2D6 for example, we constructed a database by using data from The Human Cytochrome P450 Allele Nomenclature Database[18] and cross-validating it with PharmGKB[19] and dbSNP[20].

Without loss of generality, in what follows we use CYP2D6, one of the most pharmacogenomically significant genes, to provide a detailed description of Aldy. The choice of this gene is motivated by its highly polymorphic nature and the fact that its genotyping exhibits all of the aforementioned challenges, including high allelic variability and the existence of various structural rearrangements. However, all concepts described below hold for any other gene supported by Aldy.

**Experimental data**. We have used the following data sets to evaluate Aldy's performance:

1. 96 new Coriell cell line samples spanning 32 different family trios and multiple ethnic backgrounds. They were all captured using the PGRNseq v.2 reagent and sequenced with Illumina HiSeq to an average coverage of 600×. Genotypes of the sequenced samples contain various CYP2D6 alleles, including those with multiple types of structural variation. CYP2D6 genotypes of all samples were validated with PCR-based genotyping panels. Furthermore, genotypes of nine additional ADME genes were available for 17 of these samples. We used this subset to evaluate Aldy's performance on those additional genes.

2. 137 Coriell cell line samples from GeT-RM program[21], captured using the PGRNseq v.1 reagent and sequenced with Illumina HiSeq to an average coverage of 600×[16]. Genotypes of these samples for 10 key ADME genes were inferred by Aldy and were compared with PCR-based genotyping panels[21].

3. 25 Illumina whole-genome samples from the Platinum Genome Project[22], Public CEPH 1362 Sequencing Project[23], and 1000 Genomes Project[24]. Platinum Genome samples, which include 17 individuals from the CEPH 1463 family, were sequenced using the Illumina HiSeq 2000 sequencer to an average coverage of 50×. From 1000 Genomes Project samples, we chose five samples which had (i) average coverage greater than 20× and (ii) validated genotypes available in the literature. Three CEPH 1362 samples were sequenced on HiSeq 2000 with the average coverage of 30×.

**Aldy outperforms other methods**. Aldy correctly calls CYP2D6 genotypes for all samples, significantly outperforming both Cypiripi[13], which was unable to properly genotype around 50% of the samples, and Constellation/Astrolabe[14], which misidentified around 40% of the samples (Table 1). Both Astrolabe and Cypiripi currently only support the CYP2D6 gene, and thus we used it as a means for comparison. The results in the top half of the table depict Aldy's superior performance on the 96 PGRNseq v.2 samples from data set (1). The results in the bottom half of the table summarize Aldy's predictions on 14 Illumina WGS samples (dataset (3)). In a few cases, Aldy offered more complete characterization of the genotype than the validation panels; details are discussed below. Aldy's predictions for the remaining 11 Illumina samples are available in the Supplementary Results, as we did not have validated calls for these. We note that, although we managed to run it on PGRNseq samples, Cypiripi is not designed to support such samples and its suboptimal performance is included mainly for reference purposes.

The summary of Aldy's performance on the whole set of 10 ADME genes is shown in Table 2. This table depicts Aldy's performance on (i) 137 PGRNseq v.1 samples from GeT-RM studies (dataset (2)), and on (ii) 17 PGRNseq v.2 samples for which we had available genotype validations. More than 99% of

**Table 1 Summary of CYP2D6 genotypes inferred by Aldy in comparison to other tools on the set of 32 Coriell trios, i.e., 96 PGRNseq v.2 sequenced samples, and on Illumina WGS samples, grouped by the type of the events occurring in the sample**

| PGRNseq v.2 | Aldy | Astrolabe | Cypiripi | Validation | Total |
|---|---|---|---|---|---|
| Normal | 69 | 68 | 48 | 62 | 69 |
| Deletions | 9 | 0 | 7 | 8 | 9 |
| Duplications | 6 | 0 | 4 | 5 | 6 |
| Fusions | 12 | 0 | 0 | 12 | 12 |
| Total | 96 | 68 | 59 | 87 | 96 |
| **Illumina WGS** | **Aldy** | **Astrolabe** | **Cypiripi** | **Validation*** | **Total** |
| Normal | 9 | 3 | 3 | 7 | 9 |
| Duplications | 2 | 0 | 1 | 0 | 2 |
| Fusions | 14 | 0 | 0 | 4 | 14 |
| Total | 25 | 3 | 4 | 11* | 25 |

Validation was available for 11 out of 25 Illumina WGS samples (thus the validation column, marked with *, is missing 14 samples). In a few cases, there was discrepancy between validation calls and Aldy's predictions, and after the further investigation, we confirmed that the Aldy's calls were correct (such cases occur mostly because of the lack of key SNPs in a validation kit). Further explanation is available in the discussion and Supplementary Discussion

**Table 2 Summary of genotypes for 10 ADME genes inferred by Aldy on GeT-RM samples (137 PGRNseq v.1 samples) and Coriell samples (17 PGRNseq v.2 samples)**

| PGRNseq v.1 | Matches | Improvements | Mismatches | Unknown | Total |
|---|---|---|---|---|---|
| CYP2A6 | 110 | 18 | 3 | 6 | 137 |
| CYP2C8 | 137 | 0 | 0 | 0 | 137 |
| CYP2C9 | 136 | 1 | 0 | 0 | 137 |
| CYP2C19 | 128 | 9 | 0 | 0 | 137 |
| CYP2D6 | 107 | 27 | 0 | 3 | 137 |
| CYP3A4 | 131 | 6 | 0 | 0 | 137 |
| CYP3A5 | 137 | 0 | 0 | 0 | 137 |
| CYP4F2 | 121 | 16 | 0 | 0 | 137 |
| DPYD | 79 | 56 | 0 | 2 | 137 |
| TPMT | 135 | 2 | 0 | 0 | 137 |
| Total | 1221 | 135 | 3 | 11 | 1370 |
|  | 89% | 10% | 0.2% | 0.8% | 100% |
| **PGRNseq v.2** | **Matches** | **Improvements** | **Mismatches** | **Unknown** | **Total** |
| CYP2A6 | 14 | 2 | 0 | 1 | 17 |
| CYP2C8 | 17 | 0 | 0 | 0 | 17 |
| CYP2C9 | 17 | 0 | 0 | 0 | 17 |
| CYP2C19 | 16 | 1 | 0 | 0 | 17 |
| CYP2D6 | 87 | 9 | 0 | 0 | 96 |
| CYP3A4 | 17 | 0 | 0 | 0 | 17 |
| CYP3A5 | 17 | 0 | 0 | 0 | 17 |
| CYP4F2 | 15 | 2 | 0 | 0 | 17 |
| DPYD | 9 | 8 | 0 | 0 | 17 |
| TPMT | 17 | 0 | 0 | 0 | 17 |
| Total | 226 | 22 | 0 | 1 | 249 |
|  | 91% | 8.6% | 0% | 0.4% | 100% |

"Matches" column indicates the concordance between Aldy's call and panel prediction, while "Mismatches" indicates the discordance. "Improvements" column indicates the potential improvement of Aldy's calls over the genotyping panels. Finally, "Unknown" indicates the samples for which further validation is needed to clearly call the correct genotype. Further details are available in the Supplementary Discussion

the samples were accurately genotyped by Aldy. Furthermore, Aldy found a few instances of novel major star-alleles as well; details are discussed below.

In addition to its genotyping accuracy, Aldy has very low computational overhead. In our experiments, each run required less than 10 s and fewer than 100 MB of memory even for the high-coverage PGRNseq samples. Although Astrolabe offers similar performance, it requires VCF data as an input, but generation of such data usually takes significant amounts of system resources. On the other hand, Cypiripi's running time performance significantly drops as the coverage increases and it can take up to 1 h to complete on high-coverage PGRNseq data sets.

Full results, including the complete list of obtained genotypes per sample, are available in Supplementary Results.

As can be seen from Table 1, Aldy is able to accurately identify all of CYP2D6 genotypes. Furthermore, Aldy is able to identify all complex events, especially when compared with Astrolabe (which currently does not support calling such events) and Cypiripi.

When it comes to the genotyping panels, we have found several discrepancies between Aldy's and panels' predictions. After further investigation, we concluded that genotyping panels used for validation purposes made either ambiguous or incorrect calls in these cases. Such examples include inability to detect alleles CYP2D6*35 and CYP2D6*45, which were identified as CYP2D6*2 due to the inability of validation kits to detect the discerning SNPs. Additional cases include the misidentification of the CYP2D6*15 allele, which is highly similar to the pseudogene CYP2D7. In some cases, validation kits were unclear about the exact copy number; we have cross-validated such samples with

the Illumina data and external sources to confirm our predictions. Further explanation of these cases is available in the Supplementary Discussion.

We have not observed any Mendelian inconsistencies when using Aldy on PGRNseq v.2 and Illumina WGS data. This is in sharp contrast with previous PGRNseq data analysis which relied on SNP callers to infer genotypes[16].

When it comes to Illumina WGS data, genotypes predicted by Aldy are in concordance with genotypes validated in the literature[14,21,25], as shown in Supplementary Results. Although we do not have genotype information for some members of the CEPH 1463 family, we show that our predictions are in full accord with the Mendelian laws of inheritance, as depicted in Fig. 2.

Furthermore, it should be noted that Aldy provides accurate genotype calls for other genes as well, as demonstrated in Table 2. In all samples, the accuracy rate is above 99%. A few mismatched calls are mainly due to that unlike PGRNseq v.2, the PGRNseq v.1 reagent does not target the CYP2D6 associated pseudogenes making accurate genotype calling in the presence of pseudogene-derived mutations significantly harder (Figs. 3 and 4).

Please note that majority of these samples cannot be genotyped accurately by a simple mutation tagging procedure (i.e., by testing for the existence of mutations which define the underlying star-alleles), as shown in the Table 3.

**Novel alleles**. We have detected the presence of novel major star-alleles in GeT-RM samples, namely a novel CYP2D6*10-like allele in NA17012 (*10 with c.77 G>,A), novel DPYD alleles (based on either *4, *5, or *6, with c.85 A>G in NA07357, NA10859, and

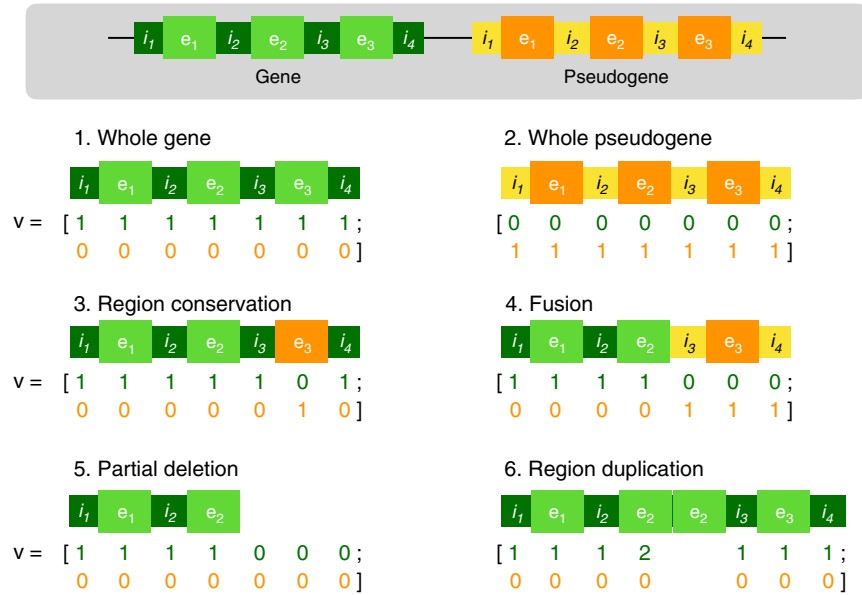

**Fig. 2** Graphical representation of the various structural configurations and their corresponding vectors **v**. Gene of interest (e.g., *CYP2D6*) is colored in green, while its pseudogene (e.g., *CYP2D7*) is colored in orange. Their typical arrangement in the genome is shown in the gray box at the top. The first two cases represent the most common configurations with whole copies of gene and pseudogene, respectively. The third case describes the conservation of pseudogene's region (in this case, exon 3) within a gene. The fourth case describes a fusion (hybrid) gene. The fifth case depicts partial deletion of the gene, while the sixth case shows how to handle cases when parts of the gene (exon 2 in this case) are duplicated

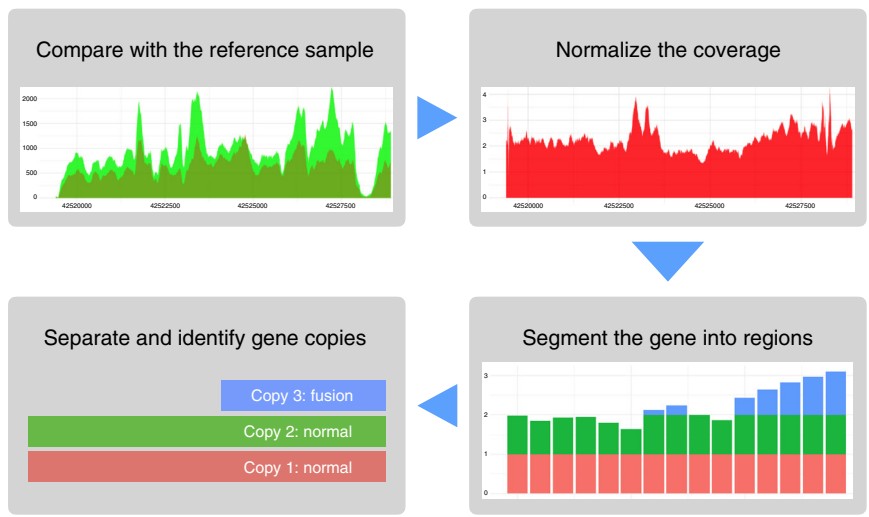

**Fig. 3** Graphical representation of structural variation detection algorithm. Initially, Aldy compares the copy number neutral input coverage with the reference sample coverage in order to establish proper depth of coverage. Only one reference sample is needed for each sequencing technology, and Aldy comes with such data by default. In the second step, the coverage of the sample is normalized (this is equivalent to establishing the function $cn_s$). The first two steps are described in Supplementary Note 1. In the third step, we segment the gene into the regions, and estimate the average copy number for each region (i.e., establish the vector **cn**). Finally, we infer the set of structural configurations (i.e., vectors **v**) which best "fit" the observed segmented coverage **cn** via integer programming

NA24027) and a novel *CYP2C19* allele in NA07439 (based on *27 with c.19153 C>T). Furthermore, we observed novel major star-alleles formed by combining variations from two different alleles: in the case of *CYP4F2*, variations from *2 and *3 allele formed a novel *CYP4F2*4* allele, and for the *DPYD* gene, a novel allele was formed by variations from both *5 and *9 alleles.

In many samples, we observe that sub-alleles detected by Aldy are not present in the online database. A similar observation was made in Raimundo et al.[26] regarding the *CYP2D6*2* family of

sub-alleles. For example, c.843 T>G, associated with all recently discovered *CYP2D6*4* sub-alleles (e.g., *4M, *4N, and *4P), is not associated with *4 alleles discovered earlier (e.g., *4A, *4B etc.). However, we found multiple samples where the evidence strongly suggests that the *4A allele contains this SNP. This implies either the incomplete characterization of *4A sub-allele, or the presence of novel *4 sub-alleles.

Since the lack of neutral SNPs can affect the accurate genotype interpretation of HTS-based tools (as already reported[14]), Aldy

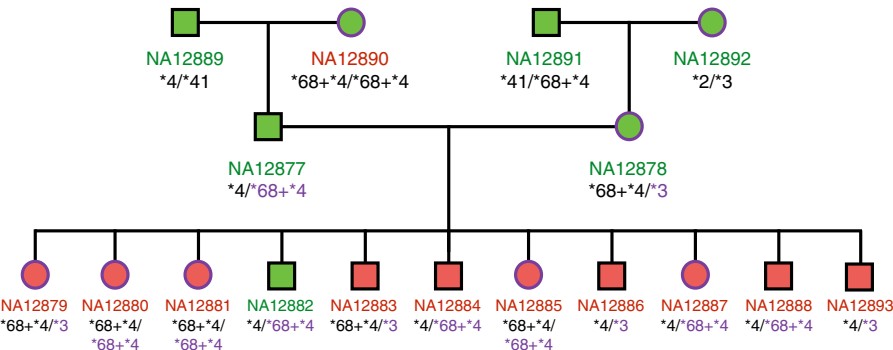

**Fig. 4** CEPH 1463 Family Tree with the Aldy genotype predictions for *CYP2D6*. Purple alleles indicate alleles inherited from mother, while black alleles from father. Red sample IDs indicate the lack of validation, while green samples the presence of validations. As can be seen, all genotypes follow Mendelian laws of inheritance

**Table 3 Summary of the samples where *CYP2D6* cannot be genotyped by simple (or single) mutation tagging (i.e., by testing for the existence of mutations that define the underlying star-alleles)**

| *CYP2D6* | PGRNseq v.2 | PGRNseq v.1 | Illumina WGS |
|---|---|---|---|
| Uniquely identifiable | 35 | 35 | 7 |
| Ambiguous | 34 | 50 | 2 |
| Subject to structural alterations (i.e., deletions, duplications, or gene fusions) | 27 | 52 | 16 |
| Total number of cases unidentifiable via mutation tagging | 61 | 102 | 18 |
| | 64% | 74% | 72% |

All samples with structural alterations (e.g., full or partial deletions, duplications, and fusions) fall into this category. The "Ambiguous" row refers to samples where mutation tagging produces ambiguous results

comes with the updated database which contains additional major star-alleles and sub-alleles believed to exist in the wild. Further studies are needed for complete characterization of those novel alleles.

## Discussion

Allelic decomposition of functionally significant genes (e.g., pharmacogenes and immunogenes) is essential for many applications, especially in clinical decision making. For example, pharmacogenes play an important role in the treatment and dosage decisions for more than 90% of all prescription drugs. Examples include *CYP2D6*, a major pharmacogene involved in the metabolism of 20–25% of clinically prescribed drugs and *CYP2A6*, a major nicotine metabolizer. Allelic decomposition of pharmacogenes is highly recommended before drug treatment decisions, even though existing methods (array based genotyping assays) are limited in scope (not covering all genes and all potential variants), can be costly and may yield false positives. Both *CYP2D6* and *CYP2A6* are located in the vicinity of highly homologous and evolutionarily related pseudogenes that facilitate formation of various structural rearrangements and copy number gains/losses, resulting in gene variants that affect a patient's drug metabolism.

Allelic decomposition of genes harboring various structural rearrangements (e.g., *CYP2D6* and *CYP2A6*) still presents a major challenge. In order to provide some level of assistance to the analysis of such structural variants, the second iteration of the PGRNseq platform (PGRNseq v.2) targets the whole genic clusters containing those genes (e.g., for *CYP2D6*, the whole 30 KB *CYP2D* cluster is targeted, including *CYP2D6* and the pseudogenes *CYP2D7* and *CYP2D8*). An additional obstacle introduced by capture panels is non-uniformity of coverage, which, despite the depth, further complicates the detection of structural rearrangements.

In this paper, we have presented the first computational tool, Aldy, to accurately infer genotypes of highly polymorphic, structurally variant but functionally important genes through the use of Illumina WGS or targeted capture sequencing data such as PGRNseq. We have applied Aldy to perform allelic decomposition of the key pharmacogenes such as *CYP2D6* and *CYP2A6*. Fast execution and low system requirements make Aldy highly suitable for clinical settings where not only accuracy but also speed is of high importance.

There are still some challenges that need to be addressed in future work. Perhaps the most important one is that in the exact characterization of novel alleles and sub-alleles, we may end up with two or more equally valid solutions. There are two possible ways to reduce the number of solutions to one, each requiring additional data: (i) if a large cohort or family information is available, use a statistical approach to identify the most likely solution; or (ii) if additional barcoded read data can be obtained, use it for phasing SNVs. We aim to expand Aldy so as to employ both types of data (when available) to break these rare occasions of symmetry.

WGS and specialized HTS platforms such are PGRNseq can help remove the last obstacles preventing the wider integration of HTS technologies in everyday clinical settings. Coupled with our accurate and fast allelic decomposition framework Aldy, these platforms can assist physicians in tailoring prescription recommendations based on a patient's genetic makeup, leading to improved and more cost-effective medical care.

## Methods

**Data sequencing**. Illumina and PGRNseq v.1 samples (data sets 2 and 3) were acquired from the public sources and the previous work[16,21–24]. We sequenced remaining 96 Coriell trio samples (data set 1) with PGRN-seq v.2 capture panel (Illumina HiSeq 2000/2500). The sequencing materials and the cell lines are

publicly available from the Coriell Institute (NHGRI Sample Repository for Human Genetic Research). All IRB and consent materials are maintained by Coriell Institute.

**Read mapping**. The common practice in genotyping is to map relevant reads to the reference genome by following a "best practices workflow"[27], usually involving a read mapper (e.g. BWA[28] or CORA[29]) and Genome Analysis Toolkit (GATK)[30]. We have used this workflow for all evaluated samples. However, Aldy is not limited to this mapping framework, and accepts any valid SAM/BAM file containing the region of the targeted gene. The GATK pipeline is recommended since it performs local indel realignment[31], which improves the detection of various small indels.

**Copy number and structural variation estimation**. *CYP2D6* (as well as many other genes) is prone to copy number variations including gene deletions and duplications, typically producing exactly two copies of a gene in a chromosome; cases in which more than two copies are present are sometimes called multiplications. It is also prone to structural variations/alterations in the form of deletions and duplications that affect only a part of the gene. Since *CYP2D6* lies in close physical proximity to highly homologous and evolutionary related pseudogene *CYP2D7*, such structural variations/alterations often lead to formation of *CYP2D6/2D7* hybrid genes, through gene fusions, where a prefix or suffix of the hybrid gene is from the pseudogene *CYP2D7*, or gene conversions, where one (or, rarely, more) non-prefix/suffix contiguous segment of the hybrid gene is from *CYP2D7* (a detailed description of known structural variations/alterations and hybrid arrangements of *CYP2D6* and *CYP2D7* can be found in Kramer et al.[32]). Note that some genes, including *CYP2D6*, have more than one pseudogene (in the case of *CYP2D6*, it is *CYP2D8*), however typically only one such pseudogene is highly homologous and difficult to distinguish. Nevertheless, the formulation below can be easily modified to accommodate multiple highly homologous pseudogenes.

Typically (gene) duplications and deletions impact a gene (such as *CYP2D6*) or a pseudogene (such as *CYP2D7*) as a whole. In the case where such a deletion or a duplication impacts only part of a gene or a pseudogene, it is called a partial gene deletion or a partial gene duplication. Since, in the literature, all structural variations/alterations resulting in the formation of hybrid genes are defined at the level of whole introns and exons[32], we assume that partial gene deletions or partial gene duplications involve one or more (contiguous sequence of) whole introns and exons. As a result, we assume that each exon and each intron of the hybrid gene originates from either *CYP2D6* or *CYP2D7* as a whole.

Let $n$ be the total number of exons and introns in a gene (as well as its highly related pseudogene—under the assumption that the number of exons and introns are conserved in the pseudogene). Now let $R = [r_1, r_2, ..., r_n, r_{n+1}, r_{n+2}, ..., r_{2n}]$ denote the sequence of introns and exons of the gene and its pseudogene, i.e., *CYP2D6* and *CYP2D7* for our case, where $r_1$ stands for the first intron and $r_2$ stands for the first exon of *CYP2D6* ($n = 19$ for *CYP2D6*). Similarly, $r_{n+1}$ and $r_{n+2}$, respectively, stand for the first intron and the first exon of *CYP2D7*. We represent each hybrid configuration of *CYP2D6/2D7* by a vector $\mathbf{v}$ of length $2n$ such that $\mathbf{v}[i] = l$ if $l$ copies of region $r_i$ are present in the configuration and $\mathbf{v}[i] = 0$ if no such copy is present. A single copy of *CYP2D6*, for the case when no structural variations are present, will be represented as a vector $\mathbf{v}$ consisting of $n$ ones followed by $n$ zeros. Analogously, this applies to a single copy of *CYP2D7* (zeros followed by ones). Note that as long as the gene of interest has $d$ well established pseudogenes with which it can hybridize, vector $\mathbf{v}$ will have $n(d+1)$ dimensions.

Aldy's first goal in this step is to derive the aggregate copy number of each exon and intron of the gene from read mappings (see below for details). Then, for each possible configuration that could be present in the sample, Aldy constructs the corresponding vector $\mathbf{v}$. Let $V = \{\mathbf{v}_1, \mathbf{v}_2, ..., \mathbf{v}_k\}$ denote the set of all such vectors. We repeat here that in theory, any vector $\mathbf{v}$ for which each of its dimensions is upper bounded by the aggregate copy number of the corresponding exon or intron, is the possible candidate vector to be included in the set $V$. However, since there are exponentially many such vectors, Aldy restricts its attention to vectors that correspond to hybrid configurations described in the database.

Aldy's main goal in this step is to find the number of whole copies of *CYP2D6* and *CYP2D7*, as well as the number of copies of each hybrid gene as well as each structural variation (more specifically, each partial gene deletion and partial gene duplication) configuration described in the database. Aldy achieves this aim by computing the set of configurations, which, collectively, match the aggregate copy number profile of each intron and exon, as closely as possible. Below, we first define how the copy number of an arbitrary genomic region (intron or exon) is estimated and then provide an exact integer linear programming formulation for achieving the goal.

In order to estimate the copy number of any region $r$ spanning positions $a, a + 1, ..., b$ of a gene or pseudogene $s$, we first calculate the normalized copy number $cn_s$ of $s$, which intuitively reflects the number of copies of $s$ at position $i$ (when the intron/exon of the gene includes ambiguously mappable positions, those positions are ignored). Details about calculating this function are provided in Supplementary Note 1 and Supplementary Figures 1 and 2. The estimated copy number (or observed coverage) of a region $r$ of $s$ is denoted as $\mathbf{cn}[r]$, and is simply

calculated as:

$$\mathbf{cn}[r] = \frac{\sum_{a \leq i \leq b} cn_s(i)}{b - a + 1}.$$

Now we formulate Aldy's goal in this step as an instance of ILP, i.e. integer linear programming (see Trick[33] for an introduction to ILP) where the goal is to find an integer linear combination $\sum_{i=1}^{k} z_i \mathbf{v}_i$ of configuration vectors from the set $V$ such that the sum

$$\sum_{r \in R} |G_r| \tag{1}$$

where

$$G_r = \mathbf{cn}[r] - \sum_{i=1}^{k} z_i \mathbf{v}_i[r] \quad \text{for each } r \in R$$

is minimized. Here, non-negative integer variables $z_i$ denote the number of times (the copy number of) a configuration described by vector $v_i$ from the database, appears in the solution (we recall that $k$ denotes the total number of possible configurations). We call this problem the Copy Number Estimation Problem (CNEP).

After finding all optimal solutions of CNEP, Aldy only reports "the most parsimonious" solution(s), i.e., those for which $\sum_{i=1}^{k} z_i$ is the minimum possible. In the case when multiple optimal solutions minimize the term $\sum_{i=1}^{k} z_i$, all will be reported in the final output of CNEP.

We illustrate the above notion of parsimony through a simple example with $n = 2$. Let the aggregate copy number vector be $\mathbf{cn} = [1111]$, and the set of potential vectors be $V = \{\mathbf{v}_1, \mathbf{v}_2, \mathbf{v}_3, \mathbf{v}_4\}$ where $\mathbf{v}_1 = [1100]$, $\mathbf{v}_2 = [0011]$, $\mathbf{v}_3 = [1000]$, and $\mathbf{v}_4 = [0100]$. Consider the following optimal solutions to this instance of CNEP: $(z_1, z_2, z_3, z_4) = (1, 1, 0, 0)$ and $(z_1, z_2, z_3, z_4) = (0, 1, 1, 1)$. The first solution is more parsimonious compared to the second one since the first solution implies the presence of the whole gene and the whole pseudogene—whereas the second solution implies the presence of two structurally altered copies of the gene itself, in addition to the whole pseudogene.

**Major star-allele identification**. One of the major goals of genotyping is to accurately predict the gene's final protein product, which is equivalent to the detection of a major star-allele of each gene copy discovered in the previous step. In order to achieve this, we first identify the set $M = \{m_1, m_2, ..., m_w\}$ of all gene-disrupting mutations (as defined in the database) detected in the sample. At this point, we discard from further consideration all major star-alleles for which at least one of their defining gene-disrupting mutations is not present in $M$. In addition, we also discard all alleles with a structural configuration that is not reported among any of the optimal solutions of CNEP (some major star-alleles of *CYP2D6* are hybrid configurations with *CYP2D7*). Let $A = \{a_1, a_2, ..., a_t\}$ denote the set of major star-alleles remaining after this filtering step and $gdm(a)$ denote the set of gene-disrupting mutations of allele $a$.

We introduce non-negative integer variables $p_1, p_2, ..., p_t$ where $p_t$ represents the number of copies of the major star-allele $a_i$ in the genotype reported in the solution. For an arbitrary mutation $m$, we define

$$E_m = \text{mutcn}(m) - \sum_{i: m \in gdm(a_i)} p_i,$$

where $\text{mutcn}(m)$ denotes the estimated copy number of $m$. Note that for a mutation $m$ (of the wild type *CPY2D6* in our case), the value of $\text{mutcn}(m)$ is obtained by normalizing the number of reads that include $m$ by the expected coverage of $m$'s locus, which is upper bounded by the normalized copy number of the region that covers $m$'s locus, since it is not necessary that all reads that are mapped to this locus include mutation $m$ —as they may originate from other copies of the gene.

We assume that the presence of gene-disrupting mutation $m$ in our sample implies that its genotype contains at least one of the major star-alleles harboring $m$. In order to enforce this, for each $m \in M$, we introduce the following constraint:

$$\sum_{i: m \in gdm(a_i)} p_i \geq 1.$$

Our objective in this step is thus to select a set of major star-alleles which most closely match the observed set $M$ of gene-disrupting mutations. More formally, we aim to minimize:

$$\sum_{m \in M} |E_m|. \tag{2}$$

We call this problem the Major Star-Allele Identification Problem (MSAIP), which we formulate/solve as an ILP. If multiple optimal solutions to the MSAIP are obtained, all of them will be processed by a final refining step.

In this step, pairs of mutation loci that are covered by a single read are optionally used to eliminate major star-alleles whose mutational compositions contradict the reads covering such mutations. More specifically, this filter considers reads that cover the locus of at least two gene-disrupting mutations, and eliminates major star-alleles which do not agree with (i.e., are not in phase with) any of the reads that cover these mutation loci.

In some rare instances, no set of major star-alleles from the database can satisfy all of the aforementioned constraints, making MSAIP infeasible. This means that the sample contains major star-alleles which are not present in the database. In such cases, an empty set is reported as the output of MSAIP and the next step (described below) is used to construct novel major star-alleles that best describe the observed set of mutations. Note that the presence of novel major star-alleles does not necessarily imply infeasibility of MSAIP; as long as the set of observed mutations can be explained with available major star-alleles from the database, Aldy will return those star-alleles as the output.

**Genotype refining**. Since certain gene-disrupting mutations are present in multiple major star-alleles, several mutations must be used to distinguish such major star-alleles. In cases where subsequent gene-disrupting mutations are "distant" (i.e., farther away than the read/fragment length), and are shared by multiple major star-alleles, it may not be possible to unambiguously identify the specific major star-alleles present in the sample.[13] In such cases, the previous step can produce multiple equally plausible solutions that "explain" the set of input reads. We resolve this ambiguity through the observation that some neutral mutations are more likely to occur within certain major star-alleles. We call the use of neutral mutations in the genotyping process the Genotype Refining Problem (GRP). We establish the final allelic decomposition of the sample by solving GRP. As described below, because the genotype refining step is highly flexible, Aldy is able to assign additional neutral mutations to each star-allele and thus establish novel star-alleles. As a result, the genotype refining step not only allows us to select the best major star-allele configuration for the sample, but also to assign each neutral mutation to its major star-allele of origin.

GRP is highly similar to MSAIP defined above. The input to GRP is a set of major star-alleles inferred in the previous step, and the goal is to "extend" each such major star-allele definition to a minor star-allele definition. As a result, the goal of GRP is to return a collection of minor star-alleles each as an extension to a unique major star-allele identified by solving MSAIP as described in the previous section. Let mut(a) denote the set of all (both gene-disrupting and neutral) mutations defining a minor star-allele a. Consider each major star-allele b identified in the major star-allele identification step and each minor star-allele a that is an extension to b (e.g., *2A is an extension of *2). Let binary variable $x_{a,b}$ indicate whether the solution of GRP "identifies" a as the "correct" extension of b. Naturally, for a given major star-allele b, $x_{a,b} = 1$ for at most one minor star-allele a. If all minor star-alleles in the sample are known/defined in the database, then the goal of GRP would be to find for each major star-allele b identified in the solution of MSAIP, the minor star-allele a for which $x_{a,b} = 1$ based on mut(a). In the case where such a solution to GRP is infeasible, we "define" a new minor star-allele a′ in the most parsimonious manner so that mut(a) ∩ mut(a′) is the maximum possible.

For each $x_{a,b}$ and for each $m \in$ mut(a), consider a binary variable $e_{a,b,m}$, which is equal to 1 only if $m \in$ mut(a′). For each $x_{a,b}$ and for each $m \notin$ mut(a) also consider a binary variable $f_{a,b,m}$, which is equal to 1 only if $m \in$ mut(a′). Now consider all mutations m, either observed in the sample, or are present in the minor allele-descriptions. The primary goal of GRP is the minimization of the (weighted) difference between $f_{a,b,m}$ and $a_{a,b,m}$ across all mutations m and all minor-major allele pairs a,b such that $x_{a,b} = 1$.

The major challenge here is thus to assign each observed mutation m to one or more of the major star-alleles identified in the previous step to obtain a possibly new minor star-allele a′ for each major star-allele b, such that $\min_{\forall a \in \text{database}} |\text{mut}(a') \cap^{\text{mut}}(a)|$ is minimized across all minor star-alleles a′ in the solution of GRP. To achieve this aim we need to ensure for each mutation m, the number of minor star-alleles a′ in the solution to GRP for which $m \in$ mut(a′) is as close as possible to mutcn(m), the estimated copy number of m (see the previous section). The difference between mutcn(m) and the number of minor star-alleles in the solution that includes m can be expressed as

$$F_m = \text{mutcn}(m) - \left[ \sum_{(a,b):m \in \text{mut}(a)} x_{a,b} e_{a,b,m} \right.$$
$$\left. + \sum_{(a,b):m \notin \text{mut}(a)} x_{a,b} f_{a,b,m} \right] \text{ for all mutations } m.$$

As a result, the objective of GRP is to simultaneously minimize $F_m$ as well as the difference between $f_{a,b,m}$ and $e_{a,b,m}$ across all mutations m. We combine these two

objectives linearly as:

$$\min \left\{ \sum_m |Fm| + \sum_{a,b} x_{a,b} \left[ \alpha \left( \sum_{m \in \text{mut}(a)} (1 - e_{a,b,m}) \right) \right. \right.$$
$$\left. \left. + \beta \left( \sum_{m \notin \text{mut}(a)} f_{a,b,m} \right) + \eta \left( \sum_{m \in M \setminus \text{mut}(a)} f_{a,b,m} \right) \right] \right\}, \quad (3)$$

where α and β, respectively, denote the penalty scores for missing and added mutations to the selected minor star-alleles from the database. In our experiments, we used α = 2 and β = 1, since the likelihood of a minor star-allele to contain a new mutation was roughly twice that of lacking a known mutation. Note that the above objective also provides a way to modify a major star-allele description: here η represents the penalty of assigning a gene-disrupting mutation to a minor star-allele (thus modifying its major allele, and thus its functional impact). The value of η is set very high (currently η = 100,000) to make sure that gene-disrupting mutations are allowed to produce novel major star-allele descriptions only if MSAIP produces no valid solutions (implying that a novel major star-allele is present in the sample). Finally, we note that $e_{a,b,m}$ is always 1 if m is a gene-disrupting mutation (i.e., we do not allow removal of gene-disrupting mutations from the minor star-allele).

We guarantee that each major star-allele associated with a minor star-allele in the solution is assigned all of its gene-disrupting mutations as follows. Let $D_a$ be the set of gene-disrupting mutations from the set mut(a). This requirement can be expressed by the following set of constraints:

$$\left( |D_a| - \sum_{m \in D_a} e_{a,b,m} \right) x_{a,b} = 0 \quad \text{for each } x_{a,b}.$$

We also ensure that no variation is "over-called" (i.e., the copy numbers of each of the minor star-alleles in the solution must be consistent with the estimated copy number of the mutations they include) through the use of additional sentinel constraints.

GRP asks us to find the set of minor star-alleles for which Eq. 3 gives the lowest score. Aldy solves GRP as a QIP (Quadratic Integer Program) and returns the minor star-alleles obtained as the final genotype. Note that Aldy solves GRP for each one of the optimal solutions for CNEP and MSAIP (which GRP takes as input). Aldy refines all such solutions, and outputs the genotype with the lowest scoring objective 3.

**Computational complexity and practical limitations**. Although particular instances of the problems described above are solvable in polynomial time (e.g., cases where configuration vectors form a unimodular matrix, as is the case of a gene not harboring any structural variations), the general case which would cover any gene is NP-hard (see Supplementary Note 2 for NP-hardness proofs for the two main components of the Aldy framework, namely copy number estimation problem—CNEP, and major star-allele prediction problem—MSAIP). The proofs use reductions from the Closest Vector Problem (CVP), a well-known NP-hard problem. Note that even approximating CVP (and, by extension, our problems) with an additive guarantee of (ln2−∈)r, for any ∈ and any r (number of regions), turns out to be NP-hard.[34]

For these reasons, we utilize state-of-the-art integer programming solvers, such as Gurobi[35] or SCIP[36], to efficiently solve our problems in practice. These solvers can identify many problem instances that are exactly solvable in reasonable time, or utilize appropriate approximation algorithms and heuristics for solving hard instances.

Regardless of the algorithm or model used for downstream analysis, Illumina-based HTS data are ultimately limited by the short read length (usually around 100 bp) and short insert size (around 300 bp). Furthermore, HTS data does not provide any information about the read's originating strand. This makes the detection of some structural rearrangements and alleles highly ambiguous, especially in homologous regions.

For example, the region between the end of exon 6 and the beginning of exon 9 in both CYP2D6 and CYP2D7 is identical. Furthermore, the size of this identical region is larger than the PGRNseq v.2's size of fragment length, implying that it is impossible to unambiguously assign reads coming from this region to the originating gene. In general, for any sequencing technology with read length R and insert size r, and given the pairwise sequence alignment for (the reference allele of) gene g and (any of) its pseudogene(s) h, reads from an identically matching region r of length greater than I + 2R cannot be unambiguously mapped to the reference genome, at least not without any additional information provided. Aldy masks each such region r during the copy number estimation step, because reads originating from r and other regions identical to r could significantly alter our estimation of the copy number of the region r. The impact of misaligned reads is clearly visible in Supplementary Fig. 1, where identically matching regions are shaded with in orange. For the purposes of fusion detection, since the sequence content of such a region in the reference allele and the related pseudogene is identical, it is impossible (and not at all relevant) to identify the exact locus of the breakpoint. As a result, masking such regions will not have any adverse effects.

Also note that there might exist multiple solutions which minimize the Eqs. 1, 2, and 3. Aldy will try to call a genotype for each such optimal solution, and select one that has the best score based on subsequent steps. However, if there are multiple optimal genotypes even after the final refining step, Aldy reports all of them as equally likely. Similarly, the underlying cause of multiple optimal solutions is again the short read length of HTS data, which prevents Aldy from precisely resolving the cases where one set of distant mutations (i.e., mutations that cannot be spanned by read pairs) describes multiple valid alleles.

Finally, due to the absence of information about the read's originating strand, it is sometimes impossible to unambiguously identify diplotype, especially in the presence of multiplications and fusions. One such example is *68+*4/*5, where both *68+*4/*5 and *68/*4 are equally likely based on HTS data analysis.

Note that all of the aforementioned limitations derive from the theoretical limitations of the sequencing technology, and not Aldy per se. Most of them can be alleviated by using technologies that provide the additional information (e.g., barcoded sequencing), and such additional information can be easily incorporated into Aldy without impacting its underlying mechanisms.

It should also be mentioned that Aldy uses the database of known fusion breakpoints to detect fusion alleles. In the very rare case of novel breakpoints (and thus a novel fusion allele), the current implementation of Aldy will select the closest breakpoint from the database as a fusion breakpoint. Novel breakpoints can be detected by extending the database to include the set of all possible breakpoints (note that in some cases, the database already includes all possible breakpoints). Similarly, Aldy relies on the database to determine whether a mutation is gene-disrupting or neutral; thus, in its current implementation (v1.0), we do not take into the account any mutation which is not part of any major or minor star-allele description present in the database.

**Data availability**. Illumina data were obtained from 1000 Genomes project (http://www.internationalgenome.org), Platinum Genomes (dbGaP phs001224), and HiSeq X CEPH sequencing performed by the National Genomics Infrastructure SciLifeLab in Uppsala, Sweden (https://export.uppmax.uu.se/a2009002/opendata/HiSeqX). All PGRNseq data are available upon request. Aldy and the sample data are available at http://aldy.csail.mit.edu.

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

## Acknowledgements

The authors wish to acknowledge the contributions of Richard Gibbs, director, and Donna M. Muzny, director of operations, together with HarshaVardhan Doddapaneni, Jianhong Hu, and the rest of the Production and Informatics/IT staff at the Baylor College of Medicine Human Genome Sequencing Center. We also acknowledge the Coriell Cell Repositories and the Center for Disease Control and Prevention's GeT-RM Program, and thank Lisa Kalman, the director of the GeT-RM Program, for providing 137 DNA samples used in this study. Furthermore, we acknowledge Natera Inc. in San Carlos, CA, and thank Swati Goyal from Natera Inc. for her help in evaluating Aldy. Finally, we would like to acknowledge Faraz Hach from Vancouver Prostate Centre for proof-reading our manuscript. C.S., B.B. and I.N. are partially supported by the US National Institutes of Health Grant GM108348 (to B.B. and C.S.). C.S. and B.B. are also partially funded by the NSERC Discovery Frontiers Grant, "Cancer Genome Collaboratory". Additional funding is provided by the Indiana University Grand Challenges Program, Precision Health Initiative (to C.S.). I.N. (partially) and S.M. (fully) are supported by Vanier Canada Graduate Scholarship. V.M.P. is supported by the IGNITE project grant (U01HG007762) and the Indiana University Health—Indiana University School of Medicine Strategic Research Initiative. X.Q. and S.S. are funded by 2U19-GM61388.

## Author contributions

I.N. and S.M. designed the study and developed the methods and the software. I.N. and M.F. performed the experiments. X.Q. evaluated the software. X.Q., L.T., M.R., V.M.P.,

T.C.S. and S.S. provided the data used in the study. I.N., S.M., S.C.S. and B.B. contributed to writing the manuscript. S.C.S., S.S. and B.B. supervised the project.

## Additional information

**Competing interests:** The authors declare no competing interests.

