## [Peer Review File · Nature Communications]

Reviewers' comments:

Reviewer #1 (Remarks to the Author):

Review of Allelic decomposition and exact genotyping of highly polymorphic and structurally variant genes, by Numanagic et al.

Most genotyping studies are limited to low frequency SNPs in nonrepetitive regions of the genomes, or genotyping polymorphic duplications that are tagged by SNPs in LD. The authors present a method to genotype pharmacogenomic relevant cytochrome genes that are subject to rearrangement. The results on the Coriell trios are promising.

While the presentation of the method is methodical, there are a number of details and extra descriptions that could be added to make it considerably more clear.

Major comments.

The manuscript does give some of the background of genotyping complex events, but from the way the results are presented, it is difficult to see the gain on previous methods or why an ILP approach is necessary. First, it is very unclear to a reader not familiar with cytochrome genotypes what the different genotypes in Table 1 represent. Some description of this is necessary. For example, please describe which haplotypes represent fusion events, copy number expansion events, etc. Next, the last step of the method - genotype refining ensures that the nonfunctional mutations in the database match those in the sample. This is quite similar to the practice of genotyping based on tagged SNVs, and relates to the Constellation method. That leads to two major points:

While it is clear that Aldy is more generalizable, it would be good to show the gain on constellation for the samples that were chosen.

This request may be more difficult, and you can ignore it, but it will make the case for the use of the method more clear: For the 3 datasets used, state how many genotypes could not be identified by a tagging SNP (a snp in that genotype and not in any others). This may arise from fusion events that do not have a SNP that distinguishes the fusion from non-fusion haplotypes.

Minor notes

Page 5 -this is a very verbose description of normalizing coverage according to a copy number invariant region. If there is space restriction this could go into a supplement.

1. Page 6, paragraph 2. What is n ?
2. Page 6, paragraph 3: k copies of CPY2D6 without hybrid genes represented as a sum of vectors - however it is stated in (1) that each vector v is in $\{0,1\}^{2n}$.
3. Please comment in this section on the effect of a copy number polymorphism that is not reflected in the database.
4. Page 9, functional mutations - if this means mutations that affect gene function, these are typically referred to as gene disrupting mutations, the other variations can be called neutral.
5. Page 9: Why is the constraint after (3) not simply a filter -- remove alleles if they contain gene disrupting variants not seen in the read set?
6. The manuscript is quite well written, but there are many sentences that are missing articles "e.g. 'the', and 'a'", for example page 5, 3rd paragraph "known that CYP2D6 genotype" -> "known that the

CYP2D6 genotype" page 5 "in the above formula leads to proper estimate" -> "... leads to a proper...".

Reviewer #2 (Remarks to the Author):

Allelic decomposition of functionally significant and highly polymorphic genes is a challenging task. Currently the short read data is rather fragmented and hence this makes this even harder. The authors described a computational method to detect the exact allelic composition of highly polymorphic genes. Here are a few comments.

1. The idea of using small variants for allelic decomposition of highly polymorphic genes is not novel. Evan Eichler's work [Science 330(6004):641-646] as well as the authors' another tool Cypiripi seem to have the same function. The same gene family was used in the Cypiripi paper. I wonder whether the authors could compare similar approaches, at least the ones themselves invented.
2. It was stated in the section 2.3 that the status of each expressed functional mutation is used to select a set of proteins. However, no transcriptome data is mentioned throughout the paper. Thus this is confusing.
3. The source code at <http://bitbucket.com/compbio/aldy> is not available at the time of reviewing this paper.

Reviewer #3 (Remarks to the Author):

The paper presents an algorithm Aldy, for determining the gene allele sequences from PGRNseq data. It uses a database of known allelic sequences to inform their analysis. The paper provides good results, but has several critical shortcomings listed below:

* There is very little discussion of the results of genes besides CYP2D6. What is the power of the algorithm to separate similar alleles? How does the performance of the algorithm depend on features of the gene, e.g. its length, number of exons, frequency of variants, distance between alleles, etc.. While the overall results of the algorithm look positive, a lack of further analysis of Table 3 makes it impossible to answer these types of questions.

* It is described how some regions of CYP2D6 need to be masked because they are identical to CDYP27. However, no rigorous method is given for determining which regions to mask. How are these regions chosen for the other genes? How should they be chosen if the user wants to apply this algorithm to genes not tested in the paper? The selection of these regions is likely crucial to the usability of the algorithm more broadly than the tested gene(s).

The algorithm detects some novel functional alleles in GeT-RM samples. However, none of these are validated. In general, there does not seem to be much biological novelty (and the authors do not seem to claim much in the Introduction). I think that for a methods paper this is not necessarily a downside, but given the journal, I think it is worth mentioning

* The authors formulate the biological question as a series of optimization problems. However, the algorithm for solving them is not described, except to say that it is an adaption of a dynamic programming algorithm for the subset-sum problem. Given that the contribution of this paper is mainly methodological, this is a critical part of the paper that is missing and necessary for evaluating it .

* The authors define Problem 1 as a generalization of the problems that they solve. They then prove that Problem 1 is NP-hard. However, this is irrelevant to the complexity of the problem they actually solve. If Problem 1 was poly-time solvable, then indeed the problems that they solve would be polynomial -- but this is not the case.

* It is not clear to me that an algorithm for subset sum has any relationship to an algorithm for Problem 1. The reduction in the proof allows one to use an algorithm for Problem 1 to solve Subset sum, but not the other way around.

* Some of the claims made in the Intro are too broad. For example, "we present Aldy, the first computation tool to perform exact allelic decomposition of any gene of interest." The authors certainly do not show this that Aldy works for any gene -- the study, at best, demonstrates this for 10 genes. As a result of such examples, parts of the paper read more like an advertisement for Aldy rather than a scientific study.

* Intro, paragraph starting with "Unfortunately, none of the above tools...": In the previous paragraph, the authors present a small selection of available tools as examples of various techniques. In this paragraph, the authors describe what these tools cannot do. However, the relevant question is what ALL the available tools cannot do, as opposed to just the hand picked examples of the previous paragraph. I suggest that the authors rewrite this paragraph to focus it on the limitations of all existing tools, rather than a few examples.

Allelic decomposition and exact genotyping of highly polymorphic and structurally variant genes

Response to Reviewers

We would like to thank the reviewers for their insightful comments and suggestions. We believe that we have addressed most of their critical points and responded to the others below; our additions and modifications to the paper are highlighted in blue.

Major changes include:

1. Complete rewrite of the Introduction section
2. Partial rewrite of the Results section, and addition of the comparison of our calls with those made by Astrolabe and Cypiripi
3. Addition of section (2.5) which describes the limitations of our method and explains the NP-hardness of underlying problems. Detailed proofs are included in Supplementary Materials.

For each reviewer comment, please find our specific response colored in blue.

Reviewer #1 (Remarks to the Author):

Most genotyping studies are limited to low frequency SNPs in nonrepetitive regions of the genomes, or genotyping polymorphic duplications that are tagged by SNPs in LD. The authors present a method to genotype pharmacogenomic relevant cytochrome genes that are subject to rearrangement. The results on the Coriell trios are promising.

While the presentation of the method is methodical, there are a number of details and extra descriptions that could be added to make it considerably more clear.

Major comments:

The manuscript does give some of the background of genotyping complex events, but from the way the results are presented, it is difficult to see the gain on previous methods or why an ILP approach is necessary. First, it is very unclear to a reader not familiar with cytochrome genotypes what the different genotypes in Table 1 represent. Some description of this is necessary. For example, please describe which haplotypes represent fusion events, copy number expansion events, etc.

We would like to thank the reviewer for pointing out these limitations. We have completely rewritten the introduction as well as the results section to highlight our improvements over previous methods and why an integer programming approach is needed. Several tables (including Table 1) have been updated according to the reviewer's suggestions.

Next, the last step of the method - genotype refining ensures that the nonfunctional mutations in the database match those in the sample. This is quite similar to the practice of genotyping based on tagged SNVs, and relates to the Constellation method.

We emphasize here that Constellation/Astrolabe is designed to do only "star-diplootyping" (calling which

pair of star-alleles—alleles defined solely by the gene-disrupting mutations present in the sample—are present for the gene *CYP2D6*, ignoring the possibility of more than two copies of *CYP2D6* in a sample, or ignoring star-alleles that involve fusions with *CYP2D7*, like *36, *68, *76, etc.). In contrast, our previous tool Cypiripi performs “star-allelotyping” of *CYP2D6*: it not only considers the possibility of more than two copies of *CYP2D6* but also calls fusion alleles with *CYP2D7*. Notably, our new method Aldy further generalizes Cypiripi, by not only handling genes other than *CYP2D6*, but also performing “exact allelotyping” rather than star-allelotyping. In particular, Aldy (based on the observation that the star-allele nomenclature is far from being complete) implicitly assembles all alleles present in the sample. In contrast, Constellation/Astrolabe (while avoiding fusions and multiple copies) and Cypiripi only aim to call star-alleles (as described by the star-allele nomenclature) that best match the sequencing data.

Coming to the reviewer’s question, Aldy not only considers the non-functional mutations in the database, but also the rare/unique non-functional mutations in the reads that do not have matching mutations in the database (and thus are not part of the star-allele nomenclature). In fact we have privately observed that the complete allele “space” for the genes such as *CYP2D6* is an order of magnitude larger than that covered by the star allele nomenclature (see Section 3.4). However because clinical pharmacogenomics (and all the published work) use the star allele nomenclature, we present Aldy’s calls in terms of star alleles in this paper- even though Aldy is capable of going beyond star alleles and report novel alleles that differ from known/star alleles.

As a result, while the genotyping step of Constellation/Astrolabe performs tagged SNV assignment for better “star-diplootyping”, Aldy goes beyond that and performs exact-allelotyping as described above. We have added a similar discussion to the manuscript.

That leads to two major points:

— While it is clear that Aldy is more generalizable, it would be good to show the gain on constellation for the samples that were chosen.

We have now included a detailed comparison of Aldy with Cypiripi and Constellation/Astrolabe in the Results section. (Note that we could not download Constellation/Astrolabe at the time of the first submission. We also would like to point that Cypiripi is not designed to handle PGRNseq data, so its performance on PGRNSeq samples is far from optimal).

In short, we found that Aldy (which called correct *CYP2D6* genotypes for all samples) significantly outperforms both Cypiripi (which was unable to properly genotype around 50% of the samples) and Constellation/Astrolabe (which mis-identified around 40% of the samples). Further details are available in the manuscript and supplementary materials.

— This request may be more difficult, and you can ignore it, but it will make the case for the use of the method more clear: For the 3 datasets used, state how many genotypes could not be identified by a tagging SNP (a snp in that genotype and not in any others). This may arise from fusion events that do not have a SNP that distinguishes the fusion from non-fusion haplotypes.

More than 70% of the known star alleles of the genes considered in our study cannot be identified by a single/tagging SNP (please see the new Table 3 in the manuscript, also shown below). Some alleles are defined by two or more SNPs that may be (individually) present in other known alleles. In addition, the star-allele nomenclature mainly covers SNPs that have an impact on the gene’s final protein product. As noted by the reviewer, synonymous SNPs and non-coding SNPs (intronic/UTR) are often ignored, even though they may have a functional impact. Aldy not only can identify the alleles that cannot be called by a single SNP (for Aldy, there is no limit on how many SNPs define a particular allele) but also has the ability to identify SNPs that are not covered by the star-allele nomenclature, CNVs, as well as fusions. We hope we have made this clearer in the updated Results section.

CYP2D6	PGRNseq v.2	PGRNseq v.1	Illumina WGS
Uniquely identifiable	35	35	7
Ambiguous	34	50	2
Subject to structural alterations (i.e. deletions, duplications or gene fusions)	27	52	16
Total fraction of cases unidentifiable via mutation tagging	61	102	18
	64%	74%	72%

Table 1: Summary of the samples where *CYP2D6* cannot be genotyped by simple (or single) mutation tagging (i.e. by testing for the existence of mutations that define the underlying star-alleles). All samples with structural alterations (e.g. full or partial deletions, duplications, and fusions) fall into this category. The "Ambiguous" row refers to samples where mutation tagging produces ambiguous results.

Minor notes

— Page 5 - this is a very verbose description of normalizing coverage according to a copy number invariant region. If there is space restriction this could go into a supplement.

As per reviewer's suggestion, we have moved this description to the supplementary materials.

— Page 6, paragraph 2. What is n ?

We have added a description of n as follows:

*For this purpose, let g and h stand for the sequences of the genes *CYP2D6* and *CYP2D7*, respectively, and assume that we split g and h into regions r_1, r_2, \dots, r_n and r'_1, r'_2, \dots, r'_n , respectively (**where n denotes the number of regions of interest in a gene**).*

— Page 6, paragraph 3: k copies of *CPY2D6* without hybrid genes represented as a sum of vectors - however it is stated in (1) that each vector v is in $\{0,1\}^{2n}$.

The following clarification has been added:

Note that the final coverage vector is not binary, but can be expressed as a sum of binary vectors.

— Please comment in this section on the effect of a copy number polymorphism that is not reflected in the database.

In the star-allele nomenclature for *CYP2D6*, there are only a handful of alleles that are reported to have multiple copies. Aldy generalizes on this by allowing any allele to have any number of copies. In case the reviewer also has fusion polymorphisms in mind, we have added the following paragraph to section 2.5:

It should be also mentioned that Aldy uses the database of known fusion breakpoints to detect fusion alleles. In the very rare case of novel breakpoints (and thus a novel fusion allele), the current implementation of Aldy will select the closest breakpoint from the database as a fusion breakpoint. Novel breakpoints can be detected by extending the database to include the set of all possible breakpoints (note that in some cases, the

database already includes all possible breakpoints).

— Page 9, functional mutations - if this means mutations that affect gene function, these are typically referred to as gene disrupting mutations, the other variations can be called neutral.

We have updated our terminology according to the reviewer's suggestions. In the new version of the paper, we use the term **gene-disrupting mutations** instead of functional mutations and the term **neutral mutations** instead of non-functional mutations.

— Page 9: Why is the constraint after (3) not simply a filter -- remove alleles if they contain gene disrupting variants not seen in the read set?

We thank the reviewer for noticing this issue. We have corrected the description as follows:

Furthermore, any major star-allele for which one or more of its defining (gene-disrupting) mutations do not have sufficient coverage/support is not considered any further.

— The manuscript is quite well written, but there are many sentences that are missing articles “e.g. ‘the’, and ‘a’”, for example page 5, 3rd paragraph “known that CYP2D6 genotype” -> “known that the CYP2D6 genotype” page 5 “in the above formula leads to proper estimate” -> “... leads to a proper...”.

We thank the reviewer for noticing this issue. We have corrected these (and many other similar) mistakes.

Reviewer #2 (Remarks to the Author):

Allelic decomposition of functionally significant and highly polymorphic genes is a challenging task. Currently the short read data is rather fragmented and hence this makes this even harder. The authors described a computational method to detect the exact allelic composition of highly polymorphic genes. Here are a few comments.

— The idea of using small variants for allelic decomposition of highly polymorphic genes is not novel. Evan Eichler's work [Science 330(6004):641-646] as well as the authors' another tool Cypiripi seem to have the same function. The same gene family was used in the Cypiripi paper. I wonder whether the authors could compare similar approaches, at least the ones themselves invented.

We thank the reviewer for pointing out the paper by Sudmant *et al.* and note that the focus of that paper is only on copy-number detection and phasing. Even though the authors use small variants to aid copy-number detection, they do not attempt to establish an exact allelic decomposition of each copy of the gene considered. We now mention the paper and its difference from our work in the Introduction as follows:

Although the work of [Sudmant10] does utilize small variants for copy number phasing, it is limited to detecting copy-number changes only, as opposed to Aldy, which can determine the exact sequence content of each copy of a gene that has been subject to structural alterations.

In addition, as we have pointed out to the first reviewer, the goal of Cypiripi is “star-allelotyping” of the gene *CYP2D6* only. Aldy generalizes Cypiripi, by not only handling genes other than *CYP2D6*, but performing “exact allelotyping” rather than star-allelotyping: while Cypiripi aims to call “star-alleles” (as described by the star-allele nomenclature) that best match the sequencing data, Aldy (based on the observation that the star-allele nomenclature is far from being complete) implicitly assembles all alleles

present in the sample and goes beyond the star-allele nomenclature.

As for the last remark, the Results section has been completely rewritten, and it now includes a comparison of Aldy with Cypiripi as well as another tool Astrolabe/Constellation. We have found that Aldy (which called correct *CYP2D6* genotypes for all samples) significantly outperforms both Cypiripi (which was unable to properly genotype around 50% of the samples) and Constellation/Astrolabe (which mis-identified around 40% of the samples). Further details are available in the manuscript and supplementary materials.

— It was stated in the section 2.3 that the status of each expressed functional mutation is used to select a set of proteins. However, no transcriptome data is mentioned throughout the paper. Thus this is confusing.

We apologize for the confusion. As the reviewer points out, Aldy uses only genome sequencing data, not transcriptomic data. We have revised the description of Aldy significantly with new (and hopefully more clear) terminology. We now use the term “gene-disrupting mutation” instead of “functional mutation”, use “neutral mutation” instead of “non-functional mutation”, and call the third step of Aldy “major star-allele identification” instead of “protein identification”. Accordingly, the sentence quoted by the reviewer above has been updated as, “the presence/absence of each gene-disrupting mutation is used to identify a set of major star-alleles”.

— The source code at <http://bitbucket.com/compbio/aldy> is not available at the time of reviewing this paper.

We apologize for this omission. We have updated the aforementioned link (which is currently <http://compbio.cs.sfu.ca/nwp-content/projects/aldy>, with username "review" and password "aldy2017"), and the new link should be working now.

Reviewer #3 (Remarks to the Author):

The paper presents an algorithm Aldy, for determining the gene allele sequences from PGRNseq data. It uses a database of known allelic sequences to inform their analysis. The paper provides good results, but has several critical shortcomings listed below:

— There is very little discussion of the results of genes besides *CYP2D6*. What is the power of the algorithm to separate similar alleles? How does the performance of the algorithm depend on features of the gene, e.g. its length, number of exons, frequency of variants, distance between alleles, etc.. While the overall results of the algorithm look positive, a lack of further analysis of Table 3 makes it impossible to answer these types of questions.

We agree with the reviewer that these parameters could be important determinants of performance. We have now added a detailed discussion on these issues, including that of the applicability of Aldy to genes beyond *CYP2D6* and its limitations in section (2.5).

It is described how some regions of *CYP2D6* need to be masked because they are identical to *CYP2D7*. However, no rigorous method is given for determining which regions to mask. How are these regions chosen for the other genes? How should they be chosen if the user wants to apply this algorithm to genes

not tested in the paper? The selection of these regions is likely crucial to the usability of the algorithm more broadly than the tested gene(s).

This is a good point and we have added a description of the masking procedure we have used to section (2.5) as follows:

In general, for any sequencing technology with read length R and insert size I , and given the pairwise sequence alignment for (the reference allele of) gene g and (any of) its pseudogene(s) h , reads from an identically matching region r of length greater than $I + 2R$ can not be unambiguously mapped to a reference genome, at least without any additional information provided.

Aldy masks each such region r during the copy number estimation step, because reads originating from r and other regions identical to r could significantly alter our estimation of the copy number of the region r . The impact of misaligned reads is clearly visible in Supplementary Figure 2, where identically matching regions are shaded in orange. For the purposes of fusion detection, since the sequence content of such a region in the reference allele and the related pseudogene is identical, it is impossible (and not at all relevant) to identify the exact locus of the breakpoint. As a result, masking such regions will not have any adverse effects.

The algorithm detects some novel functional alleles in GeT-RM samples. However, none of these are validated. In general, there does not seem to be much biological novelty (and the authors do not seem to claim much in the Introduction). I think that for a methods paper this is not necessarily a downside, but given the journal, I think it is worth mentioning.

Point well taken. The purpose of this manuscript is to provide novel algorithms and tools for adoption by the community and to leave experimental validation up to the biologists. We do however use existing data to verify our results (e.g., data from GeT-RM studies, Twist *et al.* or Fang *et al.*).

— The authors formulate the biological question as a series of optimization problems. However, the algorithm for solving them is not described, except to say that it is an adaption of a dynamic programming algorithm for the subset-sum problem. Given that the contribution of this paper is mainly methodological, this is a critical part of the paper that is missing and necessary for evaluating it .

The key methodological contribution of the paper is the combinatorial formulation of the allelic decomposition problem in the presence of structural alterations, as a combination of an instance of integer linear programming (ILP) and an instance of quadratic integer programming (QIP). We do not use a dynamic programming algorithm for the subset sum problem anywhere in Aldy. We believe the reviewer is referring to the proof that the allelic decomposition problem is NP-complete, which uses the computer science technique to demonstrate that a particular problem p (in our case the allelic decomposition problem) is very hard by showing that another problem p' (in this particular case the subset-sum problem), which is well known to be hard, would be solved in polynomial time if p could be solved in polynomial time—because of the fact that p' could be formulated as an instance of p . Although this proof demonstrates that many instances of the allelic decomposition problem are very difficult to solve (by the use of any algorithm, existing or hypothetical), thanks to our (fairly non-trivial) QIP and ILP formulations, which are put to good use by generic integer programming solvers, such as Gurobi or SCIP, we solve specific instances of the allelic decomposition problem fairly quickly. ILP and QIP are bread and butter in fields such as operations research and industrial engineering, but not so much in computational biology. Our group has been one of the pioneers in using these techniques in the field of bioinformatics, and for anyone who is interested in learning more about them we recommend several excellent tutorials such as <http://mat.gsia.cmu.edu/orclass/integer/integer.html>. We have also pointed to the tutorial in the

main manuscript and have sought to better highlight our algorithmic contribution.

— The authors define Problem 1 as a generalization of the problems that they solve. They then prove that Problem 1 is NP-hard. However, this is irrelevant to the complexity of the problem they actually solve. If Problem 1 was poly-time solvable, then indeed the problems that they solve would be polynomial -- but this is not the case.

Please see below.

— It is not clear to me that an algorithm for subset sum has any relationship to an algorithm for Problem 1. The reduction in the proof allows one to use an algorithm for Problem 1 to solve Subset sum, but not the other way around.

The reviewer is correct that the reduction shows that one can use a hypothetical algorithm for Problem 1 to solve subset sum. We use this only to prove (and this is a very standard type of a “hardness” proof used by computer scientists) that Problem 1 is very hard due to the fact that (1) the subset sum problem is already known to be hard and (2) since it can be reduced to Problem 1 in polynomial time, it must be (nearly) impossible to solve Problem 1 in polynomial time. This doesn’t mean that we use subset sum to solve Problem 1. We use a completely different technique, a combination of ILP and QIP to reformulate our problem and solve particular instances of it in a reasonable amount of time. Note that the hardness proof is provided in the paper to demonstrate how successful our ILP and QIP formulations are since they, in the instances we tested, achieve something that is theoretically proven to be difficult to achieve.

Note that this part has been completely rewritten. Supplementary Materials now include reductions from the closest vector problem (CVP) to the exact formulations of the first two problems of the Aldy framework.

— Some of the claims made in the Intro are too broad. For example, "we present Aldy, the first computation tool to perform exact allelic decomposition of any gene of interest." The authors certainly do not show this that Aldy works for any gene -- the study, at best, demonstrates this for 10 genes. As a result of such examples, parts of the paper read more like an advertisement for Aldy rather than a scientific study.

Although the proposed framework is general enough for any gene of interest, it is true that Aldy was tested on a limited set of genes of pharmacogenomic interest. Thus, we have changed this sentence to read as follows:

In order to address the aforementioned computational challenges, we present the rst framework to perform allelic decomposition of any gene of interest in HTS data. More specifically, our combinatorial framework can perform allelic decomposition of any gene that differs from the reference genome by (i) SNVs, (ii) short indels, (iii) full gene duplications or deletions (leading to copy number alteration), (iv) partial gene duplications or deletions, as well as (v) "balanced" fusions (i.e. those that preserve the structure of a gene) with highly homologous pseudogenes (the fusions can have one or more breakpoints). Our framework accomplishes this ambitious goal through a novel combinatorial optimization formulation which it uses to identify all possible combinations of genomic alterations and determine the sequence content of all copies of a gene in whole genome or targeted genome sequencing data. The practical implementation of this framework, which we named Aldy, has been used to genotype 10 of the most important pharmacogenes which regulate drug metabolism.

— Intro, paragraph starting with "Unfortunately, none of the above tools...": In the previous paragraph, the

authors present a small selection of available tools as examples of various techniques. In this paragraph, the authors describe what these tools cannot do. However, the relevant question is what ALL the available tools cannot do, as opposed to just the hand picked examples of the previous paragraph. I suggest that the authors rewrite this paragraph to focus it on the limitations of all existing tools, rather than a few examples.

We have completely rewritten the introduction, including the paragraphs mentioned by the reviewer.

Reviewers' comments:

Reviewer #1 (Remarks to the Author):

The authors have updated the manuscript in great detail according to my requests.

Reviewer #2 (Remarks to the Author):

all of my concerns have been addressed.

Reviewer #3 (Remarks to the Author):

The submission by Numanagic presents *Aldy*, an algorithm for genotyping multi-copy gene families. The paper attempts to solve an important problem. But, I found the methods section very difficult to follow and hence was not able to access the value of the contribution. I have a CS background and am comfortable with NP-hardness proofs and ILP formulations, but, on the flip side, I find abuse of notation and terminology harder to overcome than might a lay reader. I suspect that many of the issues could be fixed with a thorough proofreading of the manuscript. Nevertheless, my amount of confusion (after three re-readings) reached a point in section 2.3 where I could not continue reading the manuscript. I may be able to re-review the manuscript more fully if the writing is sufficiently improved.

Section 2.2

* First sentence: I am unaware of the term "multiplication" as applied to genes. To the best of my knowledge, this is not a standard term and should be defined.

* The second paragraph of section 2.2 is a description of prior work and is out of place in the method sections. The same can be said about the first paragraph, though to a lesser extent.

* The variable "g" is undefined in the term $\text{covg}(i)$

* In the term $\text{covg}(i)$, i is a "genic loci". In a later point (equation at bottom of page 6), it seems like it is used to refer to a single nucleotide position. But "genic loci" means to me the location of a gene. Please clarify if this means a position of a nucleotide on a chromosome, or the genomic range over which the gene is located, or something else.

* The overloading of covg to take a mutation m as a parameter is confusing. It is also not used anywhere in section 2.2.

* "...let g and h stand for the sequences of the genes...": Does this g relate to the g in covg ? The h is introduced but never used later.

* I think it would help if the fourth paragraph states at the outset that for coverage will be measured at the exon/intro level, rather than the whole gene. This is only stated as the last sentence, but it is really the key point of this paragraph.

* "...breakpoints for two splits are equivalent..." I understood what this means only on the third reading. I think it would make it easier on the reader to refer to the "splitting of the gene into regions of interest", as opposed to just saying "splits."

* What is meant by "final coverage vector" in "Note that the final coverage ..."? The vector v just presented is a "configuration" vector." Are the authors trying to relate some coverage vector that will be introduced later to this configuration vector? Or are they using the two terms interchangeably?

* The use of " m " in " $M = \{v_m\}$ " was confusing to me. If I understand the authors correctly, they are using m as just a general index of elements in the set M . That is, m is a value between 0 and k . Usually, " i " is used for such an index, which would have made it a little more intuitive. Perhaps this confusion can be resolved by just writing $M = v_1, \dots, v_k$.

* What is the meaning of the word "real" in "real data sets" at the bottom of page 6. Are there datasets which are not real (simulated?) which are not diploid? Please clarify (or maybe just remove the word "real" if it is appropriate).

* In equation 2, the variable " i " is undefined. Should it be " m " instead?

* The authors say that the goal is to find M_{opt} ...however, M_{opt} is not ever mentioned in the ILP description. It also took me awhile to understand that the v_m are the variables of the ILP. Part of the confusion is the double use of v_m for the configuration vector and for its count. I appreciate the use of the boldface to distinguish the vector from the integer, and this is technically correct. However, it adds to the notational confusion of this section. Why not use x_m to represent the counts? The second part of the confusion is the way the problem is introduced, "let v_m be a non-negative". In the following sentence, the authors could explicitly say: "We aim to find v_m that minimize"

* It is unclear to me if λ_i are 1) constants hardcoded with the value of the upper bound ($1/(k \cdot \max \dots)$) or 2) ILP variables subject to the constraint of being at most the upper bound. If it is (1), then it should be explicitly stated what λ_i is set to. Moreover, it seems that λ_i does not depend on i , so I don't see why there is not just a single constant λ . If it is (2), then it should be stated somewhere that they are ILP variables, preferably where the ILP minimization criteria is first given. Moreover, in this case, what is the biological interpretation of λ_i 's meaning?

* What does the term "most parsimonious" mean in this context? Is it simply the number of configurations that occur at least once, or is it something else? Given the confusion about the meaning of λ , I did not understand this.

* It helps me think of the second minimization term as a "tiebreaker" -- perhaps the authors can use this word to motivate it as well (if they find it appropriate).

* The following sentence does not make sense "In order to ensure that ...does not affect the set of values..."

* In this example, the CYP2D gene family has two members in the reference: CYP2D6 and CYP2D7. That is why the vector v has $2n$ elements. In general, is it correct to say that if there are i copies in the reference, that we would define a vector with $i \cdot n$ elements? It does not say this explicitly anywhere in the text, and it took me awhile before I figured it out. It would have helped if the authors could address how the size of the vector v will differ for different gene families.

* The sentence "Namely, if the above inequality..." is grammatically incorrect (for example, it is a run-on sentence)

Section 2.3

* Genetics is a rich field with much established terminology, but the authors define their own. For example, it sounds like "gene-disrupting mutation" is simply a non-synonymous mutation. The authors should use this term, or, explain how what they define is different. Second, the term "neutral mutation" has an existing definition, namely a mutation on which selection does not act or one which does not affect the fitness of an individual. The definition of the authors is inconsistent with this definition because it does not include mutations which change the protein product but do not affect fitness. Again, it might be that by "neutral mutation" the authors seemly mean a synonymous mutation.

Allelic decomposition and exact genotyping of highly polymorphic and structurally variant genes

Response to Reviewers

We would like to thank the reviewers for their insightful comments and suggestions. We believe that we have addressed most of their critical points and responded to the others below; our additions and modifications to the paper are highlighted in blue. In particular, we have rewritten the whole methods section (section 2) in order to improve clarity and flow. Changes to the main manuscript are colored in a blue font for reviewer convenience.

For each reviewer comment, please find our specific responses colored in blue below.

Reviewer #3 (Remarks to the Author):

The submission by Numanagic presents Aldy, and algorithm for genotyping multi-copy gene families. The paper attempts to solve an important problem. But, I found the methods section very difficult to follow and hence was not able to access the value of the contribution. I have a CS background and am comfortable with NP-hardness proofs and ILP formulations, but, on the flip side, I find abuse of notation and terminology harder to overcome than might a lay reader. I suspect that many of the issues could be fixed with a thorough proofreading of the manuscript. Nevertheless, my amount of confusion (after three re-readings) reached a point in section 2.3 where I could not continue reading the manuscript. I may be able to re-review the manuscript more fully if the writing is sufficiently improved.

We do apologize for the confusion. The methods section was completely rewritten, and we hope that it should be now clearer than it was.

Section 2.2

— First sentence: I am unaware of the term "multiplication" as applied to genes. To the best of my knowledge, this is not a standard term and should be defined.

We have added a short explanation of the term "multiplication" as it was previously used in the literature (e.g. Tangae *et al.*, DOI [10.1371/journal.pone.0113808](https://doi.org/10.1371/journal.pone.0113808)), and the updated text reads as follows:

CYP2D6 (as well as many other genes) is prone to copy number variations including gene deletions and duplications, typically producing exactly two copies of a gene in a chromosome; cases in which more than two copies are present are sometimes called multiplications.

— The second paragraph of section 2.2 is a description of prior work and is out of place in the method sections. The same can be said about the first paragraph, though to a lesser extent.

We have completely rewritten the section 2.2, and have removed this paragraph.

— The variable "g" is undefined in the term $\text{covg}(i)$

We thank the reviewer for noticing this. We have renamed the variable g as s , defined s , and renamed the function cov as cn . The updated text is as follows:

In order to estimate copy number of any region r spanning positions $a, a+1, \dots, b$ of a gene or pseudogene s , we first calculate the normalized copy number cn_s of s , which intuitively reflects the number of copies of s at position i .

— In the term $\text{covg}(i)$, i is a "genic loci". In a later point (equation at bottom of page 6), it seems like it is used to refer to a single nucleotide position. But "genic loci" means to me the location of a gene. Please clarify if this means a position of a nucleotide on a chromosome, or the genomic range over which the gene is located, or something else.

Term i indeed denotes the nucleotide position in a gene within a reference genome. We have resolved the above-mentioned ambiguity by removing the term "genic"; see the modified version of this sentence above (in our response to the previous point).

— The overloading of covg to take a mutation m as a parameter is confusing. It is also not used anywhere in section 2.2.

We have now introduced two distinct functions to improve clarity. One of these functions is $\text{cn}(i)$, which denotes the normalized copy number at genomic loci i , and the second function is $\text{mutcn}(m)$, which denotes the normalized copy number of a particular mutation (defined for a particular position within the gene of interest).

— "...let g and h stand for the sequences of the genes...": Does this g relate to the g in covg ? The h is introduced but never used later.

We apologize again for the confusion; we have completely rewritten this section, and removed the references to g and h . We now use specific gene names, i.e. *CYP2D6* and *CYP2D7*, to explain our process of detecting structural configurations.

— I think it would help if the fourth paragraph states at the outset that for coverage will be measured at the exon/intro level, rather than the whole gene. This is only stated as the last sentence, but it is really the key point of this paragraph.

Good point—we now mention this at the beginning of the section as follows:

*Typically (gene) duplications and deletions impact a gene (such as *CYP2D6*) or a pseudogene (such as *CYP2D7*) as a whole. In the case where such a deletion or a duplication impacts only part of a gene or a pseudogene, it is called a "partial gene deletion" or a "partial gene duplication". Since, in the literature, all structural variations/alterations resulting in the formation of hybrid genes are defined at the level of whole introns and exons (see, for example, [11]) we assume that partial gene deletions or partial gene duplications involve one or more (contiguous sequence of) whole introns and exons. As a result, we assume that each exon and each intron of the hybrid gene originates from either *CYP2D6* or *CYP2D7* as a whole.*

— "...breakpoints for two splits are equivalent..." I understood what this means only on the third reading. I think it would make it easier on the reader to refer to the "splitting of the gene into regions of interest", as opposed to just saying "splits."

We have completely rewritten this paragraph, and removed the problematic sentence.

— What is meant by "final coverage vector" in "Note that the final coverage ..."? The vector v just presented is a "configuration" vector." Are the authors trying to relate some coverage vector that will be introduced later to this configuration vector? Or are they using the two terms interchangeably?

We now use the term "observed coverage" rather than "final coverage". Accordingly, we define the "observed coverage vector" as the vector where each individual dimension represents the estimated copy number of the region associated with that dimension. We have modified the relevant paragraph as follows:

In order to estimate copy number of any region r spanning positions $a, a+1, \dots, b$ of a gene or pseudogene s , we first calculate the normalized copy number cn_s of s , which intuitively reflects the number of copies of s at position i . (when the intron/exon of the gene includes ambiguously mappable positions, those positions are ignored). Details about calculating this function are provided in Supplementary Materials. The estimated copy number (or "observed coverage") of a region r of s , is denoted as $cn[r]$, and is simply calculated as: ...

— The use of "m" in " $M = \{v_m\}$ " was confusing to me. If I understand the authors correctly, they are using m as just a general index of elements in the set M . That is, m is a value between 0 and k . Usually, "i" is used for such an index, which would have made it a little more intuitive. Perhaps this confusion can be resolved by just writing $M = v_1, \dots, v_k$.

We thank the reviewer for the suggestion. We fixed this, and also renamed M as V . The text now reads:

Let $V = \{v_1, v_2, \dots, v_k\}$ denote the set of all such vectors...

— What is the meaning of the word "real" in "real data sets" at the bottom of page 6. Are there datasets which are not real (simulated?) which are not diploid? Please clarify (or maybe just remove the word "real" if it is appropriate).

All datasets should contain two autosomes, and we agree that the word "real" was superfluous. We have removed it (and moved the related discussion to Supplementary Materials).

— In equation 2, the variable "i" is undefined. Should it be "m" instead?

We thank reviewer for noticing this. We have removed all instances of "m" in the manuscript and replaced them with "i", depending on the context.

— The authors say that the goal is to find M_{opt} ...however, M_{opt} is not ever mentioned in the ILP description. It also took me awhile to understand that the v_m are the variables of the ILP. Part of the confusion is the double use of v_m for the configuration vector and for its count. I appreciate the use of the boldface to distinguish the vector from the integer, and this is technically correct. However, it adds to the notational confusion of this section. Why not use x_m to represent the counts? The second part of the confusion is the way the problem is introduced, "let v_m be a non-negative". In the following sentence, the authors could explicitly say: "We aim to find v_m that minimize"

We apologize for the inconvenient and hard-to-follow notation. We have now updated the manuscript based on the reviewer's suggestions.

— It is unclear to me if λ_i are 1) constants hardcoded with the value of the upper bound ($1/(k \cdot \max \dots)$) or 2) ILP variables subject to the constraint of being at most the upper bound. If it is (1), then it should be explicitly stated what λ_i is set to. Moreover, it seems that λ_i does not depend on i , so I don't see why there is not just a single constant λ . If it is (2), then it should be stated somewhere that they are ILP variables, preferably where the ILP minimization criteria is first given. Moreover, in this case, what is the biological interpretation of λ_i 's meaning?

In the previous version of the manuscript, λ_i described a constant whose role was to break ties in case of multiple solutions to CNEP. (This is possible due to the observation that it is more likely to have a smaller number of structural reconfigurations than a larger number of such events—see previous studies such as Gaedigk *et al.*, DOI [10.2217/pgs.09.133](https://doi.org/10.2217/pgs.09.133)). However, we have now realized that we can achieve the same objective without introducing λ_i , and we do not use them anymore. See our modified manuscript for details.

— What does the term "most parsimonious" mean in this context? Is it simply the number of configurations that occur at least once, or is it something else? Given the confusion about the meaning of λ , I did not understand this.

The most parsimonious solution in this context is one which uses the smallest number of structural reconfigurations to explain the observed data. We have clarified it in the manuscript (see our response to the previous point).

— It helps me think of the second minimization term as a "tiebreaker" -- perhaps the authors can use this word to motivate it as well (if they find it appropriate).

Indeed, this term's main purpose was tie-breaking. Again, we refer the reviewer to the previous point.

— The following sentence does not make sense "In order to ensure that ...does not affect the set of values..."

This sentence has been removed from the manuscript since it was no longer necessary—after removal of the variables λ_i .

— In this example, the CYP2D gene family has two members in the reference: CYP2D6 and CYP2D7. That is why the vector v has $2n$ elements. In general, is it correct to say that if there are i copies in the reference, that we would define a vector with $i \cdot n$ elements? It does not say this explicitly anywhere in the text, and it took me awhile before I figured it out. It would have helped if the authors could address how the size of the vector v will differ for different gene families.

Yes, as long as the gene has i well established pseudogenes with which the gene can hybridize, vector v will have $(i+1) \cdot n$ dimensions.

— The sentence "Namely, if the above inequality..." is grammatically incorrect (for example, it is a run-on sentence)

Corrected. Thank you!

Section 2.3

— Genetics is a rich field with much established terminology, but the authors define their own. For example, it sounds like "gene-disrupting mutation" is simply a non-synonymous mutation. The authors should use this term, or, explain how what they define is different. Second, the term "neutral mutation" has an existing definition, namely a mutation on which selection does not act or one which does not affect the fitness of an individual. The definition of the authors is inconsistent with this definition because it does not include mutations which change the protein product but do not affect fitness. Again, it might be that by "neutral mutation" the authors seemly mean a synonymous mutation.

We have to disagree with the reviewer on this point. First of all, the terms "synonymous" and "non-synonymous" are used with respect to the mutations occurring in the protein-coding regions (exons), and they do not convey the full meaning as needed for our problem. In our case, many mutations which affect the protein product and/or enzyme activity occur outside the exonic regions (e.g. some of them belong to the flanking regions or introns), and are as such neither "synonymous" nor "non-synonymous". In the previous version of the manuscript, we used the terms "non-functional" and "functional", but the other reviewers pointed out that such terms are also imprecise, and that we need to use the terms "neutral" and "gene-disrupting", which should be proper for our needs. In order to satisfy all reviewers and to remove all confusion, we have now included additional footnotes explaining the 'alternative' terminology that we use.

REVIEWERS' COMMENTS:

Reviewer #3 (Remarks to the Author):

The authors have addressed many of my concerns, though there remain issues with clarity, especially in the newer text. I feel that at this late round of the submission process it would be inappropriate for me to go through and yet again give detailed feedback. At some point, it becomes the author's responsibility to get this right. I recommend that the editors make a final decision on the paper in its current state. Overall, though, I think it unlikely that there are problems with the methods that would affect the biological results of this paper. If the paper is accepted, it would benefit from having a non-expert go through it to help improve readability prior to publication.